# OpenReview forum: "Thought Purity: A Defense Framework For Chain-of-Thought Attack"
_ICLR.cc/2026/Conference — ICLR 2026 Conference Withdrawn Submission_

### Official Review · Reviewer_H5SK · 2025-10-21

**Soundness:** 3
**Presentation:** 2
**Contribution:** 2
**Rating:** 4
**Confidence:** 3

**Summary:**

This paper introduces Thought Purity, a defense framework for large reasoning models (LRMs) against Chain-of-Thought attacks. Such attacks manipulate the intermediate reasoning process of models that produce multi-step explanations. The proposed framework integrates some elements: a safety-oriented data pipeline that labels suspicious reasoning segments (`<suspect>` and `<harm>`), a reinforcement learning stage based on Group Relative Policy Optimization (GRPO) with dual reward models (Outcome Reward Model and Process Reward Model). Experiments across four reasoning datasets and three model architectures show consistent improvements against BadChain-style attacks.

**Strengths:**

1. The paper addresses reasoning-stage vulnerabilities rather than only final outputs. This focus is increasingly important as chain-of-thought reasoning becomes standard in LLMs and LRMs.
2. The experiments cover different architectures and task types.

**Weaknesses:**

1. The experiments focus entirely on the BadChain family of prompt-injection attacks. Although the authors vary injection locations and ratios, they do not test TP under different attack families.

2. The same trigger token from BadChain (`@_@`) appears in both training and evaluation. The framework may therefore partially memorize the pattern instead of learning generalized reasoning hygiene.


3. Only a few simple fine-tuning or RL baselines are compared. Existing safety defenses such as prompt sanitization or adversarial detectors are not included, which makes the relative advantage less clear.

**Questions:**

1. Have you conducted hold-out trigger or unseen target mapping experiments to evaluate generalization?
2. Could the inserted `<suspect>` or `<harm>` tags cause false positives on clean reasoning data?

---

> ### Author Response · Authors · 2025-11-13
>
> **Response to Reviewer H5SK**
>
> We appreciate your insights and respond to the key issues you raised:
>
> - **Trigger diversity**: Even though new triggers can pass through our training pipeline, their diversity and robustness must be evaluated. We will expand beyond single-symbol triggers and test different insertion positions during attacks.
> - **RL memory effects**: RL-based defenses do develop long-term memories for certain triggers. Fundamentally, however, large-model pretraining itself constitutes massive data exposure, so using fine-tuning-style methods to align or defend against harmful content is not inherently problematic.
> - **Baseline availability**: When we initiated this work, RL-based baselines for defending LRM prompt backdoors were scarce. In performance-oriented settings, users often seek to improve models in the face of degradation rather than suspect an attack—this pragmatic need motivated our baseline design, though we recognize it requires further refinement.
> - **False positives**: Mislabeling does occur but remains rare, typically when prompts already contain unusual words or special symbols.
>
> Thank you again for your advice on paper writing; we will carefully consider all of your suggestions in our subsequent work.

---

### Official Review · Reviewer_dRVx · 2025-10-29

**Soundness:** 2
**Presentation:** 2
**Contribution:** 3
**Rating:** 2
**Confidence:** 4

**Summary:**

This paper addresses the security vulnerabilities of Large Reasoning Models (LRMs), specifically focusing on their susceptibility to Chain-of-Thought Attacks (CoTA) that can subvert the model's reasoning process. The authors propose a new defense framework called "Thought Purity" (TP). This framework is built upon three main components: (1) a safety-optimized data processing pipeline that uses special tags (e.g., `<suspect>`, `<harm>`) to identify and segment malicious reasoning; (2) a reinforcement learning (RL) approach, specifically Group Relative Policy Optimization (GRPO), which is guided by both process-level and outcome-level rewards; and (3) adaptive monitoring metrics, including two new metrics proposed by the authors (Cure Rate and Reject Rate). The authors evaluate their framework on several reasoning datasets and LRM families, comparing it to a baseline "ORM-only" RL method.

**Strengths:**

1.  **Important and Timely Problem:** The paper addresses a critical and relevant issue. As models increasingly rely on complex, multi-step reasoning (like CoT), understanding and mitigating attacks against this process is a valuable area of research.
2.  **Sufficient Experimentation:** The authors have been thorough in testing their method across multiple datasets and model types, which provides a good breadth of evidence for their claims.

**Weaknesses:**

1.  **Marginal and Inconsistent Performance Gains:** This is the most significant concern. While the paper claims improvements, the empirical results shown in Table 1 are not consistently strong and the gains appear marginal. For instance, in several experiments, the defended model's ACC is not substantially improved (or is even slightly worse) than the original, and the reduction in ASR/ASRc is not always compelling. This calls into question the practical utility and robustness of the proposed framework.
2.  **High-Complexity Solution:** The TP framework introduces significant complexity. It requires a multi-stage data synthesis pipeline (generating CoT, simulating attacks, implanting special tags) followed by a full-scale RL training process. This high overhead is a major drawback, especially when the performance improvements are not overwhelmingly clear. A much simpler defense method might be preferred in practice.
3.  **Limited and Weak Baseline:** The primary baseline for comparison is an "ORM-only" model, which (as the authors show) performs very poorly, sometimes worse than the original undefended model. While this validates the authors' choice to include a PRM, it makes the TP framework's victory less impressive. The paper would be far stronger if it compared TP against other established (even if not CoTA-specific) defense methods for backdoor attacks or prompt injection, such as data filtering, simple SFT-based defenses, or other prompt-engineering-based defenses.
4.  **Overfitting to a Specific Attack:** The defense mechanism, particularly the `<harm>` tag, seems narrowly designed to counter the specific "redundant reasoning" attack pattern exemplified by the attack used in the paper. It is highly questionable whether this framework would generalize to more subtle or diverse forms of CoTA, such as attacks that slightly alter the direction of reasoning without adding an obvious, skippable "harmful" block.
5.  **Clarity and Presentation:** The paper's clarity could be improved. For example, the "Degree of Defense" (Equation 1) is formally introduced but does not seem to be explicitly measured or referenced in the experimental analysis, making its inclusion confusing. The analysis of the results could also be deeper, moving beyond a surface-level description of the metrics.

**Questions:**

Please see the weakness.

---

> ### Author Response · Authors · 2025-11-13
>
> **Response to Reviewer dRVx**
>
> Thank you for your thoughtful feedback; we address your main concerns below:
>
> - **Performance focus**: We did not “cherry-pick” datasets or outcomes; instead we trained three runs and averaged the results. RL inherently has high variance, offering both high ceilings and low floors.
> - **Computation cost**: Your point about overhead is crucial. Lasting performance improvements for large models inevitably require costs proportional to model size. While external defenses (input/output checks, post-processing) are cost-effective and widely used commercially, research into intrinsic model security remains indispensable.
> - **Integration with other defenses**: Our work provides a new component that maps backdoor inference to defensive generation. As you suggested, techniques like input detection and other SFT variants could enhance our approach, and combining them may yield stronger protections against LRM backdoors.
> - **Trigger diversity**: We will broaden the trigger set to include rare words/special tokens and study trigger positions during attacks as part of ablation and testing.
> - **Writing feedback**: Thank you for your guidance on the manuscript; we will incorporate your suggestions in future revisions.

---

### Official Review · Reviewer_JPBX · 2025-11-01

**Soundness:** 1
**Presentation:** 2
**Contribution:** 1
**Rating:** 2
**Confidence:** 4

**Summary:**

In this paper, the authors introduce Thought Purity, a defense that leverages GPT-4o-generated CoT data annotated with <suspect> and <harm> tags and trains LRMs via GRPO with dual outcome/process rewards to detect, skip, and recover from prompt-injected CoT (CoTA) attacks.

**Strengths:**

The paper focuses on reasoning models (LRMs) with CoT as a vulnerability surface rather than only LLMs and general prompt injection. That narrower scope is less studied (though there is work on CoT backdoors, as above). I do believe this scope should also be the focus of the researchers.

**Weaknesses:**

1. The underlying vulnerability (CoT prompting + backdoors/triggers) is already well documented (see BadChain, SABER, etc.). Doesn’t that make the novelty incremental rather than foundational?

2. I think the defense itself is not robust to adaptive attacks. If you are evaluating against Badchain with a fixed trigger "@_@", isn’t it a fatal flaw? An adaptive attacker can change the trigger (e.g., "cf" or natural language like "as per protocol") or vary injection timing or syntax.

3. Table 3 (Anti-TP) shows that reversing rewards already degrades performance. A real attacker would do far worse than flipping signs. Isn’t it possible for attackers to craft triggers to bypass your tags? I think the flaw here is the lack of ablation on trigger transferability.


I think the flaw here is the lack of ablation on trigger transferability.

4. Looking into the cure rate and reject rate; I see the probability of Denominator instability: If ACC-clean−ACC-attack ~= 0 (near-perfect attack), CR blows up or becomes undefined. There are no confidence intervals or statistical tests.


5. If GRPO is being used to train the model to learn the tags, then why not use PPO, DPO, or another method? Is there a valid reason why only GRPO has been chosen? Can we achieve the same results with PPO or DPO?

6. From my perspective, I only see the training with GRPO being done in addition to the insertion of <suspect>, <harm>…</harm> tags during training. But these tags are never seen at inference, correct? And the model must hallucinate them to activate defense? I believe that there is no proof that this happens reliably outside of your curated data.

7. The paper could have been stronger if the results have been further compared with other defenses; Self-Debate / Chain-of-Scrutiny (Li et al., 2024a), Paraphrasing input (Jain et al., 2024), Representation engineering (RAE) (Zou et al., 2024), Fine-tuning on refusal data (e.g., SafeRLHF)

I do believe that the paper significantly lacks novelty, which is why I provided the given score.


I do have few more questions:

Line 135: What is defense depth hierarchy

Line 152: “CoTA simulate malicious users by implanting hidden system prompts, which impose fatal constraints on the model’s output normalization”.-- Can you please elaborate more on this?


Line 311: Can you please elaborate on this: “”The OutputRL-Llama model serves as the baseline defense. The design of this model stems from people’s simple wish: if ACC decreases, then enhance its ability until it rises.”””

Line 186 to 192: The language seems a tad off. Have the claims that have been made been tested?


Minor typo in line 165: “This” should be ‘this’

**Questions:**

Please see Weaknesses

---

> ### Author Response · Authors · 2025-11-13
>
> **Response to Reviewer JPBX**
>
> We sincerely thank Reviewer JPBX for the detailed and professional comments. Your constructive suggestions for improving our work and the potential optimization avenues you proposed are highly valuable. Guided by your insights, we will address the identified shortcomings as follows:
>
> - **Trigger diversity in defense and testing**: In prompt-based backdoor attacks, even when new triggers can be extended through our training pipeline, their diversity and robustness should indeed be evaluated. We will expand from single-symbol triggers to rare words or special tokens, and analyze trigger position as part of ablation and testing.
> - **Diversity of reinforcement-learning defenses**: While GRPO currently aligns best with the LRM training ecosystem, we will also evaluate PPO- and DPO-based defenses for this attack scenario.
> - **Evaluation metrics**: Our current secondary metrics are admittedly basic and intuition-driven; incorporating confidence intervals and statistical tests will be a priority in future metric design.
> - **Comparison and integration with other defenses**: Our method maps backdoor inference to defensive generation, constituting only one component of a complete defense pipeline. Techniques such as input detection and SafeRLHF can make our approach more convincing, and we believe combining them could yield stronger defenses against LRM backdoors.
> - **RL-based defense characteristics**: RL is typically used to improve generalization by learning input–output mappings rather than hard label correspondences (unlike SFT). The hallucinations you mentioned are, in fact, a mechanism through which RL exerts its effect.
>
> Additional clarifications:
>
> - From our case-study observations we noted multiple response patterns, so we defined different levels of defense effectiveness accordingly.
> - Injecting text into the system prompt is a central vector for large-model backdoor attacks. Users often store role/task configurations in txt or json files (e.g., “You are a mathematical reasoning expert…”), separating usage from role setup—an approach common in commercial models. Prompt engineering wields more influence than fine-tuning when steering large models, which explains why jailbreak defenses evolve rapidly yet remain vulnerable.
> - When we conducted this work, there were few RL-based baselines for defending LRM prompt backdoors. In a performance-driven context, users often try to boost model capability before suspecting an attack. This straightforward motivation led us to design our baseline, which we acknowledge is incomplete and needs refinement.
>
> We appreciate your meticulous language review. Even if our work did not fully meet your expectations, you remain one of the most professional reviewers we have encountered.

---

### Note · Authors · 2025-12-02

I have read and agree with the venue's withdrawal policy on behalf of myself and my co-authors.